# Combined Pulmonary Fibrosis and Emphysema: When Scylla and Charybdis Ally

**DOI:** 10.3390/cells12091278

**Published:** 2023-04-28

**Authors:** Marija Gredic, Srikanth Karnati, Clemens Ruppert, Andreas Guenther, Sergey N. Avdeev, Djuro Kosanovic

**Affiliations:** 1Cardio-Pulmonary Institute (CPI), Universities of Giessen and Marburg Lung Center (UGMLC), Member of the German Center for Lung Research (DZL), Justus Liebig University, 35392 Giessen, Germany; 2Institute for Anatomy and Cell Biology, Julius-Maximilians-University Würzburg, 97070 Würzburg, Germany; 3UGMLC Giessen Biobank & European IPF Registry/Biobank, 35392 Giessen, Germany; 4Institute for Lung Health (ILH), 35392 Giessen, Germany; 5Lung Clinic, Evangelisches Krankenhaus Mittelhessen, 35398 Giessen, Germany; 6Department of Pulmonology, I.M. Sechenov First Moscow State Medical University (Sechenov University), 119991 Moscow, Russia

**Keywords:** CPFE, lung fibrosis, emphysema, animal models

## Abstract

Combined pulmonary fibrosis and emphysema (CPFE) is a recently recognized syndrome that, as its name indicates, involves the existence of both interstitial lung fibrosis and emphysema in one individual, and is often accompanied by pulmonary hypertension. This debilitating, progressive condition is most often encountered in males with an extensive smoking history, and is presented by dyspnea, preserved lung volumes, and contrastingly impaired gas exchange capacity. The diagnosis of the disease is based on computed tomography imaging, demonstrating the coexistence of emphysema and interstitial fibrosis in the lungs, which might be of various types and extents, in different areas of the lung and several relative positions to each other. CPFE bears high mortality and to date, specific and efficient treatment options do not exist. In this review, we will summarize current knowledge about the clinical attributes and manifestations of CPFE. Moreover, we will focus on pathophysiological and pathohistological lung phenomena and suspected etiological factors of this disease. Finally, since there is a paucity of preclinical research performed for this particular lung pathology, we will review existing animal studies and provide suggestions for the development of additional in vivo models of CPFE syndrome.

## 1. Introduction

Combined pulmonary fibrosis and emphysema (CPFE) represents an increasingly recognized, progressive lung pathophysiology. The disease is associated with strong male predominance, a lack of specific treatment options and poor prognosis. The complex pathology of this condition involves emphysematous destruction of lung parenchyma, diffuse interstitial fibrosis, changes in the composition of lung immune cells, increased production of immunomodulatory factors and the prominent remodeling of pulmonary vasculature. Such an existence of obstructive and restrictive changes in the same lungs results in unexpectedly preserved lung volumes, while in contrast gas exchange is impaired. Although the numerous preceding studies noted the coexistence of emphysema and fibrosis in the same lungs and unique physiological consequences of such an occurrence [1,2,3,4], it was Cottin et al. in 2005 who coined the term and initiated separation of this pathology into a clinical entity distinct from fibrosis and emphysema alone [5]. The versatility of phenotypes, described in a later text, and the lack of clear definitions and terminology, have prevented definite conclusions about the clinically relevant aspects of CPFE and hampered research. Until now, little is known about the etiology of the disease and the molecular mechanisms that result in two different responses to injury in the same lungs. However, it is likely that cigarette smoke and other chronic environmental exposures interact with complex genetic and developmental factors to produce this phenotype. Moreover, it is likely that the whole spectrum of different morphological phenotypes and progression patterns of CPFE exists. Additionally, a high incidence of CPFE in conditions with a strong autoimmune component prompted the hypothesis that a loss of immunological self-tolerance might play a role in this pathology. The recent efforts to correct paucities in CPFE terminology and provide a general definition of this syndrome [6] hold promise for faster developments in the future. This review will give an overview of the clinical presentation, pathophysiology, comorbidities and outcomes of CPFE. Moreover, in an effort to aid the development of preclinical research, we will focus on the known pathogenic factors, histopathological presentation and animal models that mimic the aspects of this complex pathology. The relevant literature sources analyzed in our manuscript were searched using PubMed and Google Scholar. The following keywords, alone or in different combinations, were used for the literature screening: pulmonary fibrosis, lung fibrosis, emphysema, chronic obstructive pulmonary disease, combined pulmonary fibrosis and emphysema, usual interstitial pneumonia, smoking-related interstitial lung disease, idiopathic pulmonary fibrosis, interstitial pneumonia with emphysema, emphysema with fibrosis, diffuse interstitial pneumonia with emphysema, pulmonary hypertension, computed tomography, fibrotic interstitial lung disease, fibrotic non-specific interstitial pneumonia, histology, histopathology, fibroblast foci, honeycombing, honeycomb lesions, pulmonary function, connective tissue disease, bleomycin, cigarette smoke exposure, smoke exposure, mice, animal models and transgenic mouse model. In general, the inclusion criteria were the following: (1) articles written in English language; (2) papers with appropriate study design and accurate, objective and reliable data presentation, interpretation and conclusions; (3) research and review articles that comprehensively covered the relevant topic and (4) papers that provided crucial information. The articles written in other languages and those who did not meet the inclusion criteria were excluded from further analysis. This method of the literature search is characterized by one important weakness: the neglect and ignorance of the data/findings in the papers that are not published in the English language.

## 2. Definition, Symptoms and Prevalence

CPFE is a computed tomography (CT)-defined syndrome that encompasses concurrent emphysema and interstitial pulmonary fibrosis. The syndrome is characterized by normal spirometry, impaired gas exchange, a high prevalence of pulmonary hypertension and poor survival. The vast majority of CPFE patients are males and ex- or current heavy smokers, with representation significantly higher compared to idiopathic pulmonary fibrosis (IPF) [5,7,8,9]. Almost all patients experience exertional dyspnea, recently found to originate from poor ventilatory efficiency, rather than hypoxemia as previously believed [5,10,11]. Additional symptoms and signs of the disease include cough, finger clubbing, bilateral basal crackles and occasional wheezing on auscultation [5,11,12].

The prevalence of CPFE in the general population is not known, but it has been estimated that 3.1% of asymptomatic male smokers have this condition [13]. In patients with IPF, the reported prevalence of concurrent emphysema is between 8% and 61% in different cohorts [8,14,15,16,17,18,19,20,21,22,23]. Among patients with connective tissue disease (CTD) with associated interstitial lung disease (ILD), the prevalence of emphysema was between 5 and 18% in systemic sclerosis (SSc) [24,25,26,27] and up to 48% in rheumatoid arthritis (RA) patients [27,28]. This wide range can be in part explained by different definitions of CPFE in terms of emphysema and fibrosis requirements, but also by smoking habits and genetic susceptibility in different populations.

## 3. Pathophysiology

### 3.1. Pulmonary Function Tests and Blood Gas Analysis

Long before CPFE was recognized as a separate pathology with unique pathophysiologic features, it had been noticed that the presence of concurrent emphysema in IPF patients can result in atypical, misleading pulmonary function tests [3,4,29]. Since then, it has been repeatedly confirmed that, compared to IPF patients, those with coexistent emphysema have relatively preserved lung volumes (total lung capacity (TLC), residual volume (RV) and forced vital capacity (FVC)), a generally comparable forced expiratory volume in the first second (FEV_1_), more airflow obstruction and severely depressed carbon monoxide diffusion capacity (DL_CO_) and a carbon monoxide transfer coefficient (K_CO_) [8,11,16,18,20,23,30]. In fact, in a cohort described by Cottin et al., a restrictive ventilatory defect (shown as a decrease in TLC) was found only in approximately 20% of CPFE patients [5]. Conversely, an obstructive ventilatory defect was found in only about half of patients from the same cohort despite the substantial emphysema, which illustrates the fact that compared to chronic obstructive pulmonary disease (COPD) patients, those with coexisting fibrosis have relatively preserved FEV_1_ and FEV_1_/FVC, less hyperinflation and lower DL_CO_ [5,6]. Similar conclusions were reported from the other cohorts, with the isolated low DL_CO_ being the dominant pulmonary function test pattern, followed by restriction and obstruction patterns [31]. In other words, despite the severe pathology, in some CPFE patients a reduced DL_CO_ is the only abnormal finding in pulmonary function tests, due to the counterbalance between the restrictive effects of pulmonary fibrosis and the hyperinflation caused by emphysematous lesions [5,18,32].

Blood gas analysis in CPFE patients resembles that seen in IPF patients, with resting and exercise hypoxemia, while hypercapnia rarely occurs, unlike the situation seen in severe COPD [33].

Regarding the dynamics of lung function parameters, the annual decrease in FEV1/FVC% is significantly higher in CPFE patients compared to those with IPF, but lower than that seen in COPD patients. On the other hand, the annual decrease in DLCO in CPFE patients was comparable to that observed in IPF, but significantly more pronounced than in patients suffering from COPD [7,30,34].

### 3.2. Radiographic Features

CPFE was originally individualized as a CT-defined syndrome, with unique clinical and pathophysiologic features. The diagnosis requires the presence of emphysema and interstitial pulmonary fibrosis, but a wide variety of possible appearances in chest high-resolution CT (HRCT) has been described [5,6].

#### 3.2.1. Emphysema Appearance in Computed Tomography

Inclusion criteria used by Cottin and colleagues required upper-zone predominance of emphysema, which, in regard to the relationship with reticular opacities, usually followed one of the three HRCT patterns [5,35]:emphysema lesions of the upper zones distant to fibrotic lesions of the bases;progressive transition with diffuse emphysema (centrilobular and/or bullous) and zone of transition between bullae and honeycombing;paraseptal emphysema with predominant subpleural bullae of enlarging size at the bases.

Although this requirement for the upper zone predominance limited confounding effects of emphysema, which are discussed in a further text, the current literature acknowledges that emphysema may be present in other areas of the lung, admixed with fibrosis or replaced by large cysts [6,36]. In fact, histological examination of CPFE lungs conducted by Kinoshita and colleagues suggested that the craniocaudal separation of emphysema and fibrosis is rare; instead, the coexistence or collision of fibrosis and emphysema in each lobe was noted in a large majority of samples [36]. Interestingly, Jacob et al. reported that previously described functional consequences of emphysema depend on its distribution. Namely, the study showed that only emphysema admixed with fibrosis results in the preservation of lung volumes (FVC) due to tractionally dilated airways within the fibrotic lung that prevent air trapping. In contrast, only emphysema isolated from fibrosis reduced DL_CO_, presumably because of the destruction of capillary beds in areas of isolated emphysema and the consequent reduction in blood volume within the lungs [15].

In a cohort reported by Cottin et al., centrilobular emphysema was present in the vast majority of cases and additionally, paraseptal emphysema was particularly frequent in this population (93%) [5]. Other studies corroborated the finding that paraseptal emphysema, alone or in combination with centrilobular type, occurs more frequently in CPFE than in COPD patients, thus possibly representing a hallmark of this pathology [26,37,38]. Of interest, the dominance of a paraseptal distribution of emphysema was associated with worse survival in CPFE patients [39,40]. In addition, the formation of large bullae could be observed in approximately half of the patients [5]. Of note, a recent study found that the predominant type of emphysema correlates with the type of fibrotic changes in the same lungs. Specifically, paraseptal emphysema was associated with a usual interstitial pneumonia (UIP)-HRCT pattern and a higher extent of fibrosis compared to the centrilobular pattern [41]. Crucially, the existence of thick-walled large cysts represents a unique radiologic (and histologic) feature of CPFE, absent in emphysematous and IPF lungs [42,43,44,45].

When it comes to the minimal extent of emphysema needed for the CPFE diagnosis, the original definition did not require any, but many subsequent studies imposed the minimum thresholds of lung volume affected by emphysematous changes, such as ≥5%, 10% or even 15% [46]. The ATS/ERS/JRS/ALAT task force proposes CPFE definition based on the emphysema extent being >5% of the total lung volume for research purposes, but >15% for the definition of CPFE clinical syndrome [6]. Imposing a threshold requirement for clinical purposes likely reflects the notion that the degree of emphysema affects both the annual decline rate and the prognostic significance of FVC in CPFE patients [14,23].

#### 3.2.2. Fibrotic Changes in Computed Tomography

The presence of diffuse parenchymal lung disease with fibrosis is a prerequisite for CPFE diagnosis, but a minimal extent of such changes has rarely been defined [6]. Interestingly, several studies showed that the extents of emphysema and fibrosis in CPFE lungs are negatively correlated [15,35], and that CPFE patients often have less fibrosis at initial diagnosis compared to IPF groups [3,10,18]. However, this might be a consequence of the earlier need for medical advice in CPFE patients, due to the symptoms caused by the severity of total lung involvement.

In a study by Cottin et al., the most commonly observed fibrotic changes in chest HRCT scans were honeycombing (95%), reticular opacities (87%) and traction bronchiectasis (69%). The cases with prominent ground-glass opacities were excluded, but some ground-glass opacities were nevertheless present in more than half of patients, while micronodules and airspace consolidation were rare [5,44]. Taken together, these radiological findings resulted in a definite UIP pattern in 51% of patients, strongly suggestive of UIP or fibrotic nonspecific interstitial pneumonia (fNSIP) in 34% and a complex pattern with predominant reticular opacities in the remaining cases [5]. In addition, Jankowich et al. reported a variant form of desquamative interstitial pneumonia (DIP) with extensive fibrosis, characterized by lower lobe ground-glass changes in HRCT, as a possible form of fibrosis in CPFE [47]. In a bigger group of CPFE patients, Alsumrain and colleagues reported that the largest clinicoradiologic cohort was ‘unclassifiable’ ILD, followed by a UIP/IPF. Of note, unlike Cottin et al., the authors did not exclude cases with known CTD, and those cases accounted for more than 20% of the cohort. Other classifiable ILD included suspected DIP/respiratory bronchiolitis–interstitial lung disease (RB-ILD), NSIP, chronic hypersensitivity pneumonitis (cHP) and asbestosis [31]. It is difficult to draw a conclusion about the prevalence of different patterns of interstitial pneumonia in CPFE, mainly due to the different inclusion criteria between studies. This lack of diagnostic criteria and proposed solutions are nicely reviewed elsewhere [6]. Briefly, it is now proposed that all subtypes of fibrosing ILD (fILD) with emphysema are included in CPFE populations, but with the requirement of a clear description of the fILD subtype [6].

Finally, when discussing the radiologic appearance of fibrotic changes in CPFE, one is bound to mention the confounding effects of emphysema. These include, for example, the honeycomb cyst-like appearance of emphysema in HRCT scans, in cases when it surrounds ground-glass opacities in NSIP or when it is combined with smoke-induced localized fibrotic changes in septal walls, discussed in a further text [42,48,49].

### 3.3. Histopathology

CPFE was originally defined based on clinical, physiologic and HRCT features [5], and diagnosis cannot be established on histopathological findings alone, but supportive histological features include emphysema and a pattern of fibrosis other than smoking-related interstitial fibrosis (SRIF) or Langerhans cell histiocytosis (LCH) [6]. Although the first definition of this syndrome required a predominance of emphysema in the apical part of the lungs and fibrosis of the lower zones of the lungs, most of the recent literature recognized that emphysema might be present in other areas of the lungs and even admixed with fibrosis. In addition, SRIF as well as several other (possibly overlapping) patterns of smoking-related fibrotic changes have been encountered in CPFE lung specimens and will be discussed in detail in a further text (Figure 1).

#### 3.3.1. Emphysematous Changes in Histopathology

Historically, emphysema is defined as the permanent enlargement of airspaces beyond the terminal bronchioles, accompanied by the destruction of alveolar walls, without obvious gross fibrosis [50]. Although the required absence of fibrosis might seem confusing in light of the CPFE, this limitation was likely imposed to distinguish emphysema from the revised airspaces and honeycombing in interstitial fibrosis [49]. The original definition of this disease/syndrome, given by Cottin et al. in 2005, required the upper-lobe predominance of emphysematous changes [5]. However, the most recent literature proposes that in CPFE emphysema may be present in other areas of the lungs, admixed with fibrosis or even replaced by thick-walled cysts [6,36].

As stated before, paraseptal emphysema often coexists with the centrilobular form in CPFE patients [36,51]. The occurrence of centrilobular emphysema together with fibrotic changes likely accounts for the appearance of thick-walled cystic lesions, which is a unique and common feature of CPFE [43]. These cysts (>2.5 cm in diameter) appear at the level of membranous/respiratory bronchiole, involve one or more acini and have a dense fibrous wall mainly composed of collagen with occasional fibroblastic foci. In the upper lobes, cysts are often surrounded by emphysematous tissue, while in the lower, they are adjacent to normal parenchyma and may extend to honeycombing areas. Inomata et al. reported that the appearance of these lesions in CPFE patients is associated with a greater extent of emphysema [43]. The occurrence of these cysts is thought to reflect the expansion of emphysema as it is pulled apart by the adjacent contracting fibrotic lung, therefore it has been suggested that the appropriate term for this phenomenon could be “traction emphysema” [6].

#### 3.3.2. Fibrotic Changes in Histopathology

Fibrotic changes in CPFE, although a prerequisite for a diagnosis, represent the most confusing aspect of the pathology, complicated by unclear classification and terminology. Wright and colleagues [49] attempted to classify fibrotic changes occurring together with emphysema into diffuse (such as UIP and NSIP) and localized, clinically inconsequential fibrotic events commonly occurring in the lungs of smokers.

Based on the relatively modest number of biopsy findings from CPFE lungs that have so far been described, diffuse fibrotic changes present in this pathology most often have a UIP pattern, characterized by the patchy fibrosis, honeycombing and presence of fibroblast foci [6,43,49,52]. These fibrotic lesions in emphysematous lungs are localized mainly in the lower parts of the lungs and do not show any significant difference in the type and distribution, compared to the lungs of patients with UIP alone [53].

Apart from UIP and UIP-like patterns (i.e., in the case of chronic hypersensitivity pneumonitis), NSIP, DIP as well as unclassifiable interstitial pneumonia have been encountered in histological specimens from CPFE patients [5,40,52,53].

Another scenario, clinically and pathologically very different from the abovementioned diffuse fibrosing interstitial pneumonia, has been classified by Wright as localized fibrosis that is part of emphysema or related to respiratory bronchiolitis (RB), or both [49].

In this class belongs occult interstitial fibrosis, which Katzenstein and colleagues described as a common occurrence in lobectomy samples from smokers [54] and termed SRIF. This pathological change, present in approximately half of examined specimens, is characterized by alveolar wall thickening caused by dense, relatively acellular collagen depositions, a hyalinized and ropey appearance, no or minimal accompanying inflammation, rare fibrotic foci and honeycomb changes. Emphysema is present in all cases, but fibrotic changes are also evident in non-emphysematous parts of the parenchyma. Similarly, SRIF often co-appears with other smoking-related changes, such as RB, and has been also noted in lungs with UIP [42,54].

Authors as well as others [6,49,54] acknowledge that the described pathological findings overlap with fibrotic changes previously noted by Fraig et al. and are described by Yousem in RB-ILD [55,56]. Here, changes were described as the thickening of septal walls caused by the pauci-cellular lamellae of dense eosinophilic collagen, widespread in subpleural and centrilobular areas, but most pronounced in emphysematous parts of the lungs [55]. SRIF (and, consequently, RB-ILD with fibrosis) also show similarities to airspace enlargement with fibrosis (AEF), a fibrotic change found via the gross examination of lobectomy specimens from smokers in a study by Kawabata et al. and characterized by a fibrous, and frequently hyalinized, interstitium with structural remodeling without fibroblast foci [52]. Here, it is important to note that some degree of fibrosis is a well-documented and usual occurrence in emphysematous lungs [49,57,58,59]. Similarly, the accumulation of macrophages and the delicate fibrosis of the respiratory bronchiole and surrounding parenchyma, known as RB, is virtually always present in smokers [56,60]. For that reason, some authors regard these localized fibrotic events in emphysematous lungs as an integral feature of this disease, rather than a separate pathological phenomenon [49]. Nevertheless, the current literature is uniform in the view that these fibrotic changes represent a phenomenon clearly distinct from fibrosing interstitial pneumonia, do not imply a poor prognosis and are not sufficient for CPFE diagnosis [6,49,54,55]. However, suggestions have been made that this localized fibrosis represents a significant component (or, even, a correct diagnosis) of some cases of smokers diagnosed with NSIP, DIP, unclassified interstitial pneumonia or even UIP [52,54,55]. Indeed, Otani and colleagues suggested that the combination of emphysema and SRIF that exceeds the level of resolution of the CT image has the appearance of clustered cysts with visible walls and mimics honeycombing [42]. Such a possibility would imply that some CPFE cases do not have fibrosis other than SRIF and that we need to revisit our definitions and our diagnostic criteria.

#### 3.3.3. Pulmonary Vasculopathy

Given the high prevalence of pulmonary hypertension (PH) in CPFE patients [5], it is not surprising that broad and heterogeneous, mostly moderate-to-severe vasculopathy is one of the prominent histopathological features of this disease [61,62]. Changes in pulmonary vessels encompass intimal thickening, medial hypertrophy of pulmonary arteries/arterioles and modest venopathy in both emphysematous and fibrotic areas (Figure 2) [61,62].

The extent of pulmonary vascular alterations in emphysematous and fibrous parts of CPFE lungs is comparable to that observed in pathologically changed areas of COPD and IPF lungs, respectively [61]. In addition, occasional capillary multiplication, present in emphysematous areas [61,62], as well as rare plexiform lesions [61], have been found in CPFE lungs. Strikingly and in contrast to emphysema or fibrosis alone, described pulmonary vascular changes, albeit milder, extended to the preserved areas of CPFE lungs [61,62].

### 3.4. Inflammatory Cells and Mediators

With regard to the profiles of inflammatory cells in bronchoalveolar lavage (BAL), CPFE pathology seems indistinguishable from IPF. The differential cell pattern in BAL fluid from CPFE patients is characterized by a reduced proportion of macrophages compared to the healthy lungs, increased neutrophils and occasionally elevated lymphocytes and eosinophils [5,22,63]. Similarly, the quantification of numerous immunomodulatory molecules in the diseased lung tissue suggested that the inflammatory profiles of IPF and CPFE are indistinguishable but differ significantly from emphysema. Surprisingly, the same study found that the levels of the inflammatory proteins in the emphysematous, fibrotic and mixed disease regions of the CPFE lungs are essentially the same [64].

However, an investigation of cytokine profiles in BAL fluid by Tasaka et al. revealed that levels of chemokine (C-X-C motif) ligand (CXCL)-5 and CXCL-8 are significantly elevated in CPFE patients compared to those with fibrosis alone. Additionally, the study suggested that levels of CXCL-8 correlate with neutrophil accumulation in the alveolar space and impairment of DL_CO_ [22]. The authors proposed that this difference may be attributed to the finding that CXCL-8 levels in BAL fluid distinguish between subjects with or without emphysema among current smokers [22,65]. Interestingly, the previously mentioned study from Cornwell et al. found no difference in CXCL-8 expression between lung tissues with IPF, emphysema or CPFE pathology [64].

Several studies investigated the presence of different ILD biomarkers in CPFE and their usefulness in differentiating between this pathology and IPF. The previously mentioned study by Tasaka and colleagues found that the presence of emphysema in patients with fibrosis is associated with decreased levels of lactate dehydrogenase (LDH) and increased C-reactive protein (CRP) in serum, whereas there were no significant differences in the amount of ILD biomarkers’ surfactant protein (SP)-D and Krebs von den Lungen (KL)-6 [22]. Cornwell and colleagues also reported no difference in SP-D expression between emphysema, IPF and CPFE, based on the analysis of the diseased lung tissue [64]. On the other hand, another study found that levels of KL-6 and cytokeratin-19 fragments (*CYFRA 21-*1**) in the blood of CPFE patients were the same as in those with emphysema and significantly reduced compared to IPF pathology [66]. Curiously, in yet another cohort, KL-6, SP-D as well as club cell protein (CC)-16 levels in CPFE patients were significantly higher compared to the group with emphysema alone [67]. These contradicting reports might be the consequence of different inclusion criteria and/or severity of the disease in tested CPFE patients, since KL-6 and SP-D, and in particular their combined product (KL-6 × SP-D), were found to be good indicators for an estimation of the degree of fibrosis in CPFE [68].

Intriguingly, some of the changes that would be expected based on animal studies have never been found. For example, levels of transforming growth factor (TGF)-β in CPFE patients did not differ significantly from those measured in either the blood or BAL of emphysema and IPF patients, although there was a significant increase in IPF compared to the emphysema group [22,66]. Similarly, tumor necrosis factor (TNF)-α levels in lung tissue and BAL were not different between CPFE patients and those with fibrosis alone [22,53,64]. Finally, the matrix metalloproteinase (MMP)-9 amount in the blood of CPFE patients did not differ from that measured in IPF and emphysema groups [66]. Nevertheless, emphysematous areas in CPFE lungs were characterized by stronger MMP-9 expression compared to emphysematous lungs, whereas fibroblastic foci in CPFE and IPF lungs expressed comparable levels of this enzyme [53].

### 3.5. Comorbidities

Clinically, most relevant comorbidities in CPFE are PH and lung cancer [6]. PH, currently defined as a mean pulmonary arterial pressure (mPAP) above 20 mmHg (with pulmonary vascular resistance ≥2 Wood Units for pre-capillary PH), is a common complication of COPD and IPF [69,70,71]. Therefore, it is not surprising that elevated mPAP is a common occurrence in CPFE syndrome. However, the estimation of the PH prevalence in this clinical entity is, as many aspects discussed above, complicated, with unclear diagnostic criteria and a vast heterogeneity of definitions used in the literature. PH has been reported in 15–55% of CPFE patients [6], with some authors reporting a higher prevalence compared to fibrosis alone [5,26] and the existence of PH in virtually all CPFE patients [17,72], and others failing to confirm this finding [9,10,21,24]. The prominent remodeling of pulmonary vasculature, described in detail in a further text, would seem to corroborate the worse phenotype in CPFE compared to emphysema and IPF alone [61,62]. Moreover, it has been reported that the presence of emphysema in IPF is associated with worse PH, in terms of significantly higher mPAP estimated via echocardiography [18]. Along these lines, Mejía and colleagues found that the severity of PH correlates with the extent of emphysema in CPFE patients [17]. Interestingly, Sugino et al. followed a cohort for one year and reported that, while the baseline values of mPAP estimated via echocardiography did not differ between IPF patients with and without emphysema, the annual increase of mPAP was significantly higher in the CPFE group [21]. Unsurprisingly, several studies indicated that severe PH is a predictor of poor survival in patients suffering from this syndrome [17,21,73].

Numerous studies suggested a higher prevalence of lung cancer in CPFE compared to IPF alone, and this was corroborated by a few recent meta-analyses [74,75], although individual studies differ in their conclusions [6]. CPFE is associated with a seven-fold higher incidence and worse survival of lung cancer patients compared to those without this condition [75]. The type of cancer most commonly found in CPFE lungs is squamous cell carcinoma [75], in contrast to the general epidemiology where adenocarcinoma is more prevalent [76]. However, this might not be surprising, as squamous cell carcinomas are more strongly associated with heavy smoking and are generally predominant in male smokers [76,77].

## 4. Pathogenesis

Pathological mechanisms involved in the development and progression of CPFE are insufficiently understood. The coexistence of these two smoking-related pathologies might be occurring accidentally, by fibrosis being superimposed on emphysematous lungs or vice versa [49,53]. However, it has been also proposed that CPFE is a distinct entity with a shared pathogenic mechanism [5,33,78]. Moreover, little is known about the sequence of pathogenic events. It has been speculated that emphysema in fibrotic lungs is the consequence of the traction of the upper zones of the lung by fibrotic lower parts. Still, there is little evidence for such a scenario, as radiographic fibrotic changes rarely precede the onset of emphysema [5]. On the other hand, the occurrence of fibrotic changes after emphysema diagnosis has been reported in the literature [79]. Until now, environmental triggers such as smoke exposure, CTD-ILDs, and genetic factors have been associated with increased susceptibility for CPFE development and will be discussed in detail in a further text.

### 4.1. Smoking and Occupational Exposures

Cottin et al. coined the term CPFE based on the observations made in a cohort of 61 patients, all of whom were current or former cigarette smokers. Since then, smoking remained the best-documented risk factor for the development of this syndrome. Numerous reports indicate that CPFE patients are almost exclusively former or current heavy smokers, with a more extensive smoking history and greater mean smoking pack years compared to those suffering from IPF [7,16,19,72,80]. Reviewing the available literature in 2012, Jankowich found that 98% of reported CPFE cases (with recorded smoking history) were current or ex-smokers [81], and a similarly high prevalence of smokers was reported in numerous studies published since then [7,9,21,31,43]. As cigarette consumption is a known risk for both IPF [82,83] and COPD [84,85], it is not surprising that the combination of these conditions is also more often found in smokers. However, there is a prevailing view that environmental exposures, such as cigarette smoking and other noxious insults discussed in the further text, interact with genetic factors to produce this combined pathology in susceptible individuals [6,81,86].

Apart from smoking, other environmental and occupational exposures have been recognized as the risk factors for developing co-existing fibrotic and emphysematous changes, independently or together with smoking. Interestingly, in a cohort consisting mainly of non-smoking female patients chronically exposed to indoor biomass-burning smoke, pathological lung changes included emphysema, but also reticulonodular, honeycombing and ground-glass opacities [87]. In accordance with typical CPFE presentation, the coexistence of obstructive and restrictive changes in the same individual resulted in preserved lung volumes. In addition, Daniil et al. found a significant portion of CPFE cases with paraseptal emphysema in a cohort of pulmonary fibrosis patients, all of whom were male smokers and farmers chronically exposed to agrochemical compounds [88]. Similarly, it is well known that occupational exposure induces pneumoconiosis in coal workers, which in severe cases leads to progressive massive fibrosis (PMF) [89,90], but can also result in fibrotic pattern that mimics IPF [91]. In addition, chronic exposure to mineral dusts predisposes coal workers to the development of emphysema [92,93,94,95], and co-occurrence of these two coal dust-related pathologies can result in a typical CPFE pathology, with prominent emphysema, lower lobe-predominant interstitial fibrosis and severe diffusion abnormalities [91,96]. Farmer’s lung disease and chronic bird breeder’s disease, forms of hypersensitivity pneumonitis, are both associated with the development of emphysema and CPFE [97,98,99,100]. Finally, exposure to talc in a tire factory worker, iron dust in a welder and rare earths in a movie projectionist have also been linked to the development of CPFE, supporting the possible role of environmental exposures in the pathogenesis of this disease [81,101,102,103,104].

### 4.2. Connective Tissue Diseases and Other Autoimmune Disorders

Investigations in patients with CTD-ILD, such as SSc-associated ILD (SSc-ILD) and rheumatoid arthritis-associated ILD (RA-ILD), demonstrated that CPFE may also be observed in an etiologic context different than an exposure to cigarette smoke [27]. Several studies demonstrated that among SSc-ILD patients, emphysema occurs not only among male heavy smokers but also among females, never-smokers, and smokers with a low pack-year history, especially when compared to the previously reported average tobacco consumption in “idiopathic” (smoking-related) CPFE [24,25,26,105,106]. In contrast to the conflicting data from cohorts with unknown or mixed CPFE etiologies, in the existing literature, the occurrence of emphysema on top of interstitial lung disease in SSc patients is uniformly described as associated with higher morbidity and worse survival [24,25,26]. Similarly, in RA-ILD patients, emphysema was found in approximately half of the cohort consisting of current/ex-smokers with a relatively low pack-year smoking history [28]. Strikingly, among never-smokers with RA-ILD, emphysema and, consequently, CPFE, were found in 27% of cases, where they were associated with worse outcomes [107]. Interestingly, in accordance with a report by Cottin et al. that one-third of CPFE patients had elevated nonspecific antinuclear antibodies despite the absence of overt CTD, a recent study demonstrated that compared to patients with IPF alone, CPFE patients have an increased incidence of nonspecific, but also highly specific autoantibodies for microscopic polyangiitis (MP) [5,108]. Authors suggested that occult autoimmune disorder with multi-organ manifestations, but different treatment options and more favorable prognosis, may reside inconspicuously under CPFE diagnosis [108,109]. Indeed, the elevation of autoantibodies specific for microscopic polyangiitis, as well as the condition itself, have been described as possible causes of CPFE pathology, and the possibility that CPFE occurs as a first pathological change and precedes MP has also been described [109,110,111].

While in the “idiopathic” forms of CPFE the pathophysiological mechanisms remain elusive, in the case of CTD, it is believed that mechanisms intrinsic to the disease itself, such as autoimmunity and inflammation, are in play [26]. Indeed, evidence was provided that chronic and deregulated inflammation in the lung, as well as loss of immune tolerance and autoimmunity, contribute to the development of emphysema and COPD [112,113,114]. In addition, a suggestion has been made that peripheral vasculopathy, a common finding in SSc-ILD patients, may participate in the destruction of the fibrously thickened alveolar walls [105].

### 4.3. Genetic Factors

Some cases of CPFE may also have a genetic component. Multiple genetic variants, especially in the genes coding for components of the telomere maintenance system and surfactant metabolism, are associated with a higher susceptibility to emphysema and/or fibrosis in carrying individuals or families. In addition, studies in animal models which we describe in a further text, have provided several candidate genes and signaling pathways potentially involved in the pathogenesis of CPFE, but their relevance for the human disease remains to be investigated.

Replicative senescence due to telomere shortening is an interesting topic with regard to the pathogenesis of CPFE. Accelerated telomere attrition and mutations in the components of telomerase have been associated with higher susceptibility for IPF [115,116,117,118]. In fact, IPF is the most common manifestation of telomere-mediated disorders, and mutations in telomere and telomerase genes might explain one-third of familial IPF cases [119,120]. Similar, albeit less uniform reports, suggest that the shortening of telomeres is also accelerated in COPD patients and that emphysema is a recurrent manifestation of telomere syndromes in smokers [121,122,123]. Therefore, it is not surprising that CPFE is one of the possible phenotypic outcomes in susceptible individuals carrying deleterious mutations in essential telomerase genes, TERT and TR [122,124,125]. Of note, all of the reported CPFE cases in families carrying mutations in telomerase components were former or current smokers, while non-smokers were more likely to develop fibrosis alone [122,124,125]. Such a situation implies the importance of environmental exposures and possibly, co-exposures, in the development of CPFE in genetically susceptible individuals. Interestingly, a recent study found that a TERT polymorphism carrying higher susceptibility to CPFE was associated with a reduced risk for CPFE while not associated with COPD [125]. Moreover, some recent studies suggested that telomere shortening is a relevant mechanism for CPFE susceptibility in a subset of patients beyond those with mutations in the telomerase core components [126]. For example, in a study by Stanley et al., a loss-of-function mutation in the RNA biogenesis factor NAF1 resulted in telomere shortening and thus predisposed carriers to CPFE [126]. Similarly, Guzmán-Vargas and colleagues reported the association between CPFE susceptibility and polymorphism in DSP gene coding for desmoplakin, a protein that has recently been linked to telomere maintenance [125,127].

Mutations in gene encoding surfactant protein C (SP-C) have been previously described as a frequent cause of familial interstitial pneumonia [128,129,130,131]. Interestingly, in addition to ground-glass opacities and septal wall thickening, HRCT scans in patients carrying a mutation in this gene revealed multiple large cysts [131,132]. More recently, Cottin et al. provided evidence that SFTPC mutation can result in CPFE syndrome, even in a non-smoker [133]. Similarly, mutations in ABCA3, a gene encoding a membrane transporter critically involved in surfactant metabolism, have been linked to neonatal and pediatric ILD, and more recently, adult pulmonary fibrosis [134,135,136]. Again, the resulting disease presented on HRCT with ground-glass opacities and large thick-walled cysts [132,136]. In parallel, two studies reported that mutations in this gene can also result in CPFE pathology, corroborating the role of surfactant metabolism in CPFE [137,138].

TGF-β signaling plays a prominent role in tissue repair, a process whose deregulation is a common denominator for emphysema and fibrosis [139]. In rodent models, overexpression of TGF-β leads to pulmonary fibrosis, whereas mice deficient in TGF-β signal transmission spontaneously develop emphysema, likely through the upregulation of MMP-9 [139,140,141]. An investigation of genetic polymorphisms in MMP-9 and TGF-β1 genes found that the T allele frequency in the promoter of MMP-9 (C-1562T) was higher in CPFE than in IPF patients and comparable to that observed in the emphysema group [142]. Interestingly, Rogliani and colleagues reported that interstitial fibroblasts present in the areas of parenchymal destruction in emphysema/UIP expressed MMP-2, -7, -9 and membrane-type (MT)-1-MMP with higher, although variable, intensity than in emphysema, which suggested the possible role of these proteases in accelerating the process of destruction and remodeling of emphysema in CPFE patients [53]. Regarding the TGF-β polymorphisms, Xu et al. found no significant difference in the TGF-β genotype distribution between the CPFE group and two other groups, although the difference was noted between IPF and emphysema patients, with the T allele frequency of TGF-β1 in T869C being higher in patients with emphysema than in those with IPF [142]. Although the polymorphisms in other components of the TGF-β signaling cascade might be interesting to investigate in the context of CPFE susceptibility, to our best knowledge such studies have not been performed to date.

Apart from mutations affecting components of the telomerase complex, surfactant metabolism and TGF-β signaling, CPFE has been associated with a mutation in the gene encoding the receptor for advanced glycation end-products (RAGE), previously associated with other lung diseases, including COPD and IPF [143]. Prolidase deficiency, a clinically heterogeneous disorder, may contribute to the development of pulmonary fibrosis and emphysema, at least in combination with detrimental environmental factors [144]. Finally, research in preclinical animal models yielded several candidate genes and pathways with possible involvement in CPFE pathology. These are, for example, genes encoding TNF-α and interleukin (IL)-13, whose overexpressions induce CPFE-like phenotype in mice [145]. However, less is known about their role in human disease. TNF-α polymorphisms have been linked to increased susceptibility to COPD [146,147,148] and different forms of interstitial fibrosis [12,149]. Of interest, TNF-308 A allele is a risk factor for developing both COPD and IPF [12,147,148,149], and rs1800925 T allele of the IL-13 gene has been associated with increased risk of COPD and worse lung function in IPF patients [150,151]. Future research is needed to address the relevance of these genetic variants for CPFE pathology.

## 5. Animal Models of CPFE

### 5.1. Combined Cigarette Smoke Exposure and Bleomycin Lung Injury in Mice

Numerous studies investigated pathological changes caused by the combination of cigarette smoke exposure and bleomycin injury in mice and reported a palette of different outcomes, ranging from severe (exclusive) emphysema phenotype to aggravated interstitial fibrosis, including even the protective effects of smoke exposure against fibrosis development. Such variability can be attributed to the different experimental designs, with single versus repetitive applications and assorted dosages of bleomycin, diverse smoking regimens of variable duration as well as several possible sequences of these injuries (Table 1). Thus, it is very difficult to deduce the most suitable combination of smoke exposure and bleomycin administration for the modeling of CPFE, and a study that will systematically compare these different options reported in the literature is still needed.

For example, when 28 days-long nose-only exposure of rats to cigarette smoke (4 h/day) is initiated together with single intratracheal instillation of (2.5 mg/kg) bleomycin, the resulting lung pathology is the interstitial fibrosis, similar in appearance but more severe compared to that observed in the bleomycin-only group. Of note, this relatively short exposure to smoke only did not produce emphysematous changes either alone or in combination with bleomycin [152]. Similarly, 40 days-long smoke exposure aggravated pulmonary fibrosis caused by a set of five intraperitoneal bleomycin injections (40 mg/kg at days 1, 5, 8, 11 and 15). Intriguingly, the reported regimen of smoke exposure (sidestream smoke of 12 cigarettes administered for 1 h per day, 5 days per week) produced fibrotic rather than emphysematous changes in the lung of experimental animals [153].

In contrast, an old report from Takada et al. describes the situation where a single intratracheal dose of bleomycin (5 mg/kg) given to hamsters in the middle of a 2-month long smoke exposure period induced emphysematous changes in this animal model [154]. However, whereas histological findings were consistent with emphysema development, lung function measurements indicated that the extensibility of smoke- and bleomycin-challenged lungs tended to decrease, contrary to the situation normally reported for emphysematous lungs [70,165]. Such an effect could be easily explained by the co-existence of restrictive changes, like in the case of human CPFE patients. Importantly, the used regimen of smoke exposure alone did not cause emphysema in experimental animals.

Recently, Cass and colleagues reported that alterations in pulmonary macrophage populations, induced by 15 weeks of cigarette smoke exposure (twelve cigarettes, twice per day), prevented fibrotic changes following intratracheal bleomycin instillation [155].

Finally, one of the possible pathological outcomes from the challenge of rodent lungs with cigarette smoke and bleomycin is combined pulmonary fibrosis and emphysema. Kulshrestha and colleagues exposed rats to smoke for up to three months (four cigarettes during 1 h per day), in combination with one intratracheal bleomycin instillation (7 U/kg) at the beginning of the 7th week and induced heterogeneous parenchymal remodeling with alternating areas of emphysema and fibrosis [156]. In accordance with the literature, smoke exposure alone induced progressive reactive oxygen species (ROS) accumulation, interstitial inflammation and emphysema [166], while bleomycin application induced prominent fibrotic changes [167]. As expected, both bleomycin and smoke alone, as well as the combination of these two injuries, resulted in pulmonary vascular remodeling, thus modeling vascular alterations seen in respective human pathologies [61,70,167,168]. Similarly, Zhang et al. reported histological changes consistent with both fibrosis and emphysema development in mice who were given one intratracheal instillation of bleomycin (2 mg/kg) after thirteen months of daily exposure to high concentrations of sidestream smoke for short periods of time [157]. Of note, Zhang and colleagues demonstrated that another fibrotic insult, murine gammaherpesvirus 68 (MHV-68), can produce CPFE pathology in mice if applied intratracheally after the described regimen of smoke exposure. However, the authors noted that these two models differ in some pathophysiological aspects: the fibrotic score, BAL fluid cytokine levels and lung macrophage and neutrophil infiltration were more pronounced in the bleomycin group, whereas the rate of alveolar epithelial cell apoptosis was higher in MHV-68-treated mice.

Nevertheless, the examples given above effectively illustrate the vast variability between smoke exposure regimens used in preclinical research [70,169], which could pose a problem for the replication of the CPFE model reported by these two groups. Thus, slight modifications and adjustments of the protocol, specific for the smoke exposure system, will likely be needed in each laboratory aiming to establish this animal model in the future.

To the best of our knowledge, the combination of smoke with other fibrosis-inducing substances such as amiodarone, silica or asbestos, has not been described in the literature.

### 5.2. Transgenic Mouse Models with Co-Existing Emphysema and Fibrosis

To date, several genetic modifications in animal models have been found to produce varying degrees of emphysema and fibrosis (Table 1). Perhaps the most commonly cited example of CPFE-like pathology in transgenic mice is the overexpression of TNF-α in the lungs. Indeed, Lundblad and colleagues found several lung function abnormalities, as well as airspace enlargement, loss of small airspaces, increased collagen and thickened pleural septa in aged transgenic mice compared to littermate controls [145]. Interestingly, this finding reconciled several earlier reports of TNF-α overexpression-induced pathology. Namely, previous studies found either progressive pulmonary fibrosis in young animals [170] or severe inflammation, emphysema, pulmonary hypertension and bronchiolitis without evidence of lung fibrosis [171]. Curiously, it has also been found that overexpression of TNF-α in the lung confers resistance to the induction of fibrosis via the administration of bleomycin or TGF-β [172,173].

Furthermore, overexpression of platelet-derived growth factor (PDGF)-B in the lung results in emphysematous lesions, focal fibrosis and lung inflammation. However, this transgenic line does not appear to be a suitable model for CPFE, since the airspace enlargement likely results from developmental abnormalities, and there is a significant phenotype variation among the individual mice from the same lineage [158].

Another transgenic mouse line reported to develop both emphysematous and fibrotic lesions is characterized by inducible overexpression of interleukin (IL)-13 in a lung-specific manner [159]. However, while emphysema is indeed present, the model is characterized by marked inflammation, reversible mucus cell metaplasia and fibrotic changes that are mainly peribronchial and peribronchiolar. Therefore, this model does not correspond to the situation seen in human CPFE but rather mimics some prominent features of asthma and COPD pathology [159].

Similarly, the absence of SP-C leads to a phenotype consisting of emphysema, monocytic infiltration, epithelial cell dysplasia and areas of interstitial thickening with increased α-smooth muscle actin expression and intensive trichrome staining [160]. Taken together, knockout of SP-C in mice produces a phenotype with the combination of emphysema and interstitial pneumonitis, although fibrotic changes are not as prominent as in bleomycin-challenged rodents or some other transgenic lines such as conditional Nedd4-2 lung-specific knockout mice [174]. Interestingly, as stated in the previous section, the phenotype of CPFE has been recently associated with familial SFTPC mutation in a non-smoker [133].

Finally, Collum and colleagues recently reported a model replicating the most important hallmarks of CPFE pathology, namely emphysema, pulmonary fibrosis and PH. Their experimental setup included supplementing adenosine deaminase-deficient (Ada^−/−^) mice with polyethylene glycol-modified ADA that allowed them to live normally until week 24, and then gradually reducing supplementation to allow the accumulation of excess extracellular adenosine and consequent lung injury. Such a protocol resulted in airspace enlargement, reduced arterial oxygenation, macrophage accumulation, deposition of collagen, marked remodeling of pulmonary vessels and PH [161,175,176]. Treatment with hyaluronan synthesis inhibitor could counteract fibrotic but not emphysematous changes in this model. Of note, authors also found increased levels of hyaluronan synthase and adenosine A2B receptor in the lower lobes of CPFE lungs, corroborating the possible relevance of this pathway for human pathology.

To the best of our knowledge, exposure of transgenic animals spontaneously developing emphysema or fibrosis, with the insult resulting in another pathological feature, i.e., bleomycin or cigarette smoke exposure, respectively, has never been undertaken. Such a combination might mimic the effects of detrimental environmental exposures in genetically susceptible individuals and could be a useful tool for the preclinical studies of this pathology.

### 5.3. Combined Emphysema and Fibrosis in Other Animal Species

Reports of CPFE-like pathologies in other animal species are scarce in the literature. Histopathological changes in dog lungs produced by chronic exposure to cigarette smoke comprise admixed emphysema and fibrosis, without evident pulmonary vascular remodeling [162,177]. Of note, in this particular study, dogs were inhaling cigarette smoke directly through the tracheostomy tube and cigarette consumption gradually increased to the level of human heavy smokers. Thus, this does not represent an example of spontaneous development of CPFE in dogs, but rather faithful replication of lung pathology seen in human heavy smokers in another animal species.

Apart from these old reports, combined emphysema and fibrosis were found to spontaneously develop in horses [163,164]. The majority of the reported cases were diagnosed with COPD, but had increased collagen deposition and in severe cases, all established signs of fibrosis including honeycombing [164].

## 6. Management and Prognosis

Unfortunately, CPFE is an amalgam of several pathologies with limited therapeutic options, focused on the alleviation of symptoms and deceleration of disease progression [69,70,178]. There are no clear guidelines for the clinical practice and treatment of CPFE. Currently employed treatment options, usually extrapolated from the evidence available for COPD and IPF pathologies, are thoroughly reviewed elsewhere [6]. Briefly, general recommendations include smoking cessation and avoidance of other inhalation exposures, as well as vaccination against influenza, Pneumococcus and COVID-19, unless contraindicated [6,179]. Supplemental oxygen therapy is used in the context of hypoxemia, at rest or during exercise [6,179]. Pulmonary rehabilitation, a standard of care in emphysema and increasingly used in fILD, is recommended to most CPFE patients. However, retrospective analysis of data from an admittedly small cohort suggested that patients with COPD might derive greater benefits than those with CPFE, at least from the relatively short periods of inpatient pulmonary rehabilitation [180]. Regarding pharmacologic therapies commonly used in COPD and IPF patients, those might have beneficial effects in CPFE patients [181,182] but should be tailored to the individual, with careful recognition of the patient’s phenotype [6,179]. In general, the only curative option currently available for CPFE is lung transplantation, shown to be feasible and coupled with favorable outcomes in CPFE patients [183]. However, it was reported that compared to the IPF patients, those with CPFE are more likely to develop primary graft dysfunction (grade 3), acute cellular regression (grade ≥ 2) and chronic allograft dysfunction after lung transplantation. Nevertheless, despite these distinctive outcomes, the study reported comparable survival of IPF and CPFE patients after transplantation [184]. When it comes to the treatment of comorbidities, the approach to the management of lung cancer in CPFE is generally similar to other populations. On the other hand, there is limited evidence to support beneficial effects of PH-specific therapies in the patients with group 3 PH in general [70,185]. In line with this, currently available controlled data mostly do not show any survival improvement with the use of PH therapies in CPFE patients [6,73,186]. The lack of significant effect of PH treatment in this disease was suggested early by Cottin and colleagues and further supported by the finding that the high extent of emphysema is predisposing IPF patients to poor outcomes in the trial with soluble guanylate cyclase stimulator, riociguat. Similarly, inhaled prostacyclin analog, treprostinil, did not significantly mitigate disease progression in CPFE patients [73,187,188]. However, Tanabe at al. demonstrated better survival of CPFE patients with severe PH when treated with phosphodiesterase-5 (PDE5) inhibitors [189]. In addition, case reports suggested improved pulmonary artery compliance upon treatment with PDE5 inhibitor tadalafil and long-term improvement of clinical symptoms following the treatment with endothelin-1 receptor antagonist, ambrisentan, in CPFE patients with PH [190,191]. Taken together, these data urge for further evaluation of PH-specific therapies in CPFE patients, particularly in cases with preserved spirometry and out-of-proportion PH [6].

CPFE syndrome is characterized by high mortality, which in different cohorts ranged between 2.1 and 8.5 years [5,7,8,17], with a 5-year survival between 38 and 55% [5,192]. Needless to emphasize, the presence of fibrosis negatively impacts the survival of emphysema patients [193]. However, it is still unclear whether the presence of concomitant emphysema changes the outcome of fILD, with some studies reporting significantly worse [9,17,21], better [8], or equal [7,15,16,18,30,194] survival of CPFE patients compared to those with fibrosis alone. These contradictory reports might be explained by attrition bias, but also by large heterogeneity of selection criteria and consequent differences in the relative extent of fibrosis and emphysema, and different proportions of non-IPF ILD cases in cohorts, which are characterized by better survival [6,40]. Notably, when fibrosis was quantified, CPFE had worse survival compared to IPF patients with a similar extent of fibrosis [9,21], and similar survival as those with more prominent fibrotic changes [11,18,195]. In other words, it seems likely that the total extent of the disease on HRCT, calculated as the sum of fibrosis and emphysema extent, is a determinant of the mortality in CPFE [11,195].

Finally, as stated earlier, the presence of lung cancer and PH is associated with poor prognosis [5,17,21,192,196]. Other reported predictors of mortality in CPFE are DL_CO_, composite physiologic index (CPI) as a measure of fibrosis extent, and age [6,7,31,53,197].

## 7. Conclusions

CPFE is an increasingly recognized syndrome, with unique pathophysiological features and a wide variety of radiologic and histologic appearances. The disease is debilitating, progressive and still untreatable. Further preclinical studies and clinical trials are needed to elucidate pathophysiological mechanisms and improve the management of CPFE.

## Figures and Tables

**Figure 1 cells-12-01278-f001:**
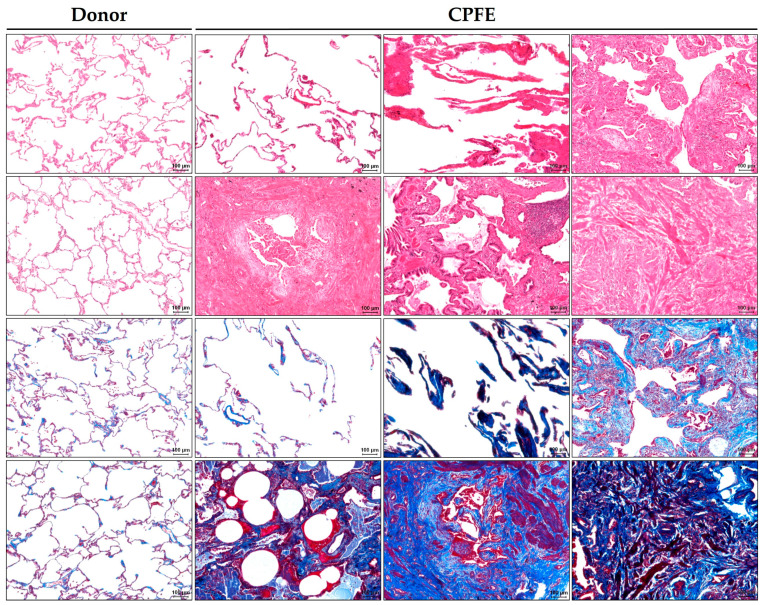
Histopathological findings in combined pulmonary fibrosis and emphysema (CPFE) lungs. The donor and CPFE lungs were stained with HE (first two rows) and trichrome (last two rows) staining. In comparison to the donor’s lungs, CPFE pathology is characterized by emphysema, occasionally accompanied by localized fibrous thickening of alveolar septa known as smoking-related interstitial fibrosis (SRIF), interstitial fibrosis with numerous fibrotic foci, fibrotic regions and honeycombing. Scale bars = 100 µm.

**Figure 2 cells-12-01278-f002:**
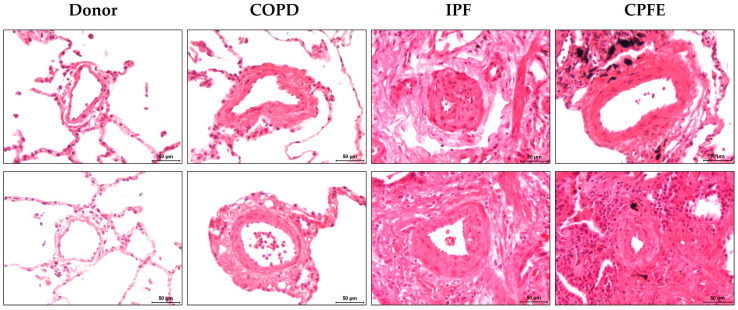
Pulmonary vascular remodeling in the pathology of combined pulmonary fibrosis and emphysema (CPFE). The lungs of donors and patients with chronic obstructive pulmonary disease (COPD), idiopathic pulmonary fibrosis (IPF) and CPFE were stained with HE. Remodeled pulmonary vessels in these diseases were shown in comparison to each other and in contrast to the “healthy” donor vessels. Scale bars = 50 µm.

**Table 1 cells-12-01278-t001:** Animal models of CPFE.

Combined Cigarette Smoke Exposure and Bleomycin Injury in Rodents					
Study	Animal Species	Smoke Exposure	Bleomycin Injury	Emphysema Pathology	Fibrosis Pathology
Sung-Moo et al. [152]	Rats	Nose-only exposure to 100, 200 or 300 μg total particulate matter/L, 4 h/day for 28 days	Single intratracheal dose of 2.5 mg/kg bleomycin	None	Stronger than in the bleomycin-only group, with more prominent neutrophilic infiltration and BALF cytokine concentrations
Li-Ling et al. [153]	Mice	Exposure to sidestream smoke from 12 cigarettes for 1 h/day, 5 days/week for 40 days	5 intraperitoneal injections of 40 mg/kg bleomycin at days 1, 5, 8, 11 and 15	None	Stronger than in the bleomycin-only group
Takada et al. [154]	Hamsters	3 sets of 4 cigarettes/5 min (separated by 5 min room air flow), twice per day, 5 days/week, for 2 months	Single intratracheal dose of 5 mg/kg bleomycin, 25 days after the start of smoke exposure	Histologically evident, more prominent than in the smoke-only group. Lack of changes in lung function	None reported, but lack of lung function changes might indicate the coexistence of obstructive and restrictive pattern
Cass et al. [155]	M = Mice	2 sets of 12 cigarettes (3R4F without filters), 5 days/week for 15 weeks	Single intratracheal dose of 0.05 U/mouse at the beginning of the 12th week	Changes in lung elastance, atelectasis and inspiratory capacity	Prevented by smoke, no significant changes in the percentage of myofibroblasts and in collagen staining
Kulshrestha et al. [156]	Rats	Sidestream smoke from 4 cigarettes for 1 h/day, 5 days/week for 12 weeks	Single intratracheal dose of 7 U/kg at the beginning of the 7th week	Histologically evident, same as in smoke-only group	Histologically evident, alternating with emphysematous areas. Pulmonary vascular remodeling is also present
Wan-Guang et al. [157]	Mice	Sidestream smoke from 10 cigarettes (at a concentration of 1000 mg/mm^3^), for 1 h/twice a day for 13 months, 5 days/week	Single intratracheal dose of 2 mg/kg bleomycin, 28 days before sacrifice	Histologically evident, same as in smoke-only group	Histologically evident
**Combined cigarette smoke exposure and mouse gamma herpes virus (MHV)—68 infections**					
**Study**	**Animal species**	**Smoke exposure**	**MHV infection**	**Emphysema** **pathology**	**Fibrosis** **pathology**
Wan-Guang et al. [157]	Mice	Sidestream smoke from 10 cigarettes (at a concentration of 1000 mg/mm^3^), for 1 h/twice a day for 13 months, 5 days/week	Single intratracheal dose of 1 × 10^5^ plaque-forming units of MHV-68, 28 days prior to sacrifice	Histologically evident, same as in smoke-only group	Histologically evident, albeit milder than in the bleomycin group
**Transgenic mouse models**					
**Study**	**Animal species**	**Genetic modification**		**Emphysema** **pathology**	**Fibrosis** **pathology**
Lundblad et al. [145]	Aged mice	TNF-α overexpression under the control of SP-C promotor		Complex lung function changes and increased lung volumes; reduction of the number of small airspaces on histology	Septal wall thickening, increased collagen
Hoyle et al. [158]	Aged mice	PDGF-B overexpression under the control of SP-C promotor		Existing throughout the lung, likely the consequence of congenital abnormalities	Confined to focal areas, inflammation present
Fulkerson et al. [159]	Mice	Inducible IL-13 overexpression under the control of CC10 promotor		Histologically confirmed	Peribronchial and peribronchiolar collagen deposition, inflammation also present
Glasser et al. [160]	Aged mice	SPC knockout		Histologically confirmed	Increased collagen deposition and α-smooth muscle actin staining, monocytic infiltration
Collum et al. [161]	Mice	ADA knockout mice supplemented with polyethylene glycol (PEG)-modified ADA until week 24, after which PEG-ADA was gradually reduced over 9 weeks. Physiological readouts were performed in week 38		Histologically confirmed	Histologically confirmed. Macrophage accumulation and pulmonary vascular remodeling are present
**Other animal species**					
**Study**	**Animal species**	**Experimental intervention**		**Emphysema** **pathology**	**Fibrosis** **pathology**
Frasca et al. [162]	Dogs	Long-term smoke exposure through a tracheostomy tube, various concentrations and durations (715–1156 days)		Histologically confirmed	Fibrous thickening of alveolar walls
Picavet et al. [163], Kaup et al. [164]	Horses	None		Histologically confirmed	Extensive peribronchial and peribronchiolar fibrosis, desquamative changes, patchy fibrotic changes with a pronounced increase in collagen fibers and honeycombing

TNF-α: tumor necrosis factor–α. SP-C: Surfactant protein-C. IL-13: Interleukin 13. CC10: Club cell protein 10. PDGF-B: platelet-derived growth factor B. ADA: Adenosine deaminase.

## Data Availability

Data are available from corresponding author upon reasonable request.

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
