# Peer review of "Combined Pulmonary Fibrosis and Emphysema: When Scylla and Charybdis Ally"

_cells, 2023, doi:10.3390/cells12091278_

Round 1

Reviewer 1 Report

In the paper entitled "Combined pulmonary fibrosis and emphysema: when Scylla 2 and Charybdis ally" by Gredic et al, the Authors provide a comprehensive overview of CPFE, from a clinical, radiological and pathological point of view associated with the collection of experimental works based on animal models. The review is well written and of interest to clinicians involved in the management of this disease.

These are my minor concerns:

-The paper lacks an iconographic section. It would be interesting to integrate the radiological and histological images of an emblematic case of CPFE

-I would also add a final section dedicated to what is reported in the literature regarding the transplantation of lungs affected by CPFE (such as: PMID: 27214130).

Author Response

Reviewer 1.

Q1: In the paper entitled "Combined pulmonary fibrosis and emphysema: when Scylla 2 and Charybdis ally" by Gredic et al, the Authors provide a comprehensive overview of CPFE, from a clinical, radiological and pathological point of view associated with the collection of experimental works based on animal models. The review is well written and of interest to clinicians involved in the management of this disease.

R1: We thank the reviewer for the nice comments about our manuscript.

Q2: These are my minor concerns:

-The paper lacks an iconographic section. It would be interesting to integrate the radiological and histological images of an emblematic case of CPFE

R2: We agree with the reviewer that the manuscript would benefit from the iconographic section. Accordingly, we added two figures (pages 6 and 7) depicting commonly encountered pathological changes in parenchyma and pulmonary vasculature in CPFE lungs.

Q3: -I would also add a final section dedicated to what is reported in the literature regarding the transplantation of lungs affected by CPFE (such as: PMID: 27214130).

R3: We thank reviewer for this suggestion. Unfortunately, we believe that mistake has occurred in the copy-paste process, since paper with the suggested PMID is focused on the recurrent reproductive failure. Nevertheless, we searched the literature and added the following text (Section 6: Management and prognosis; lines 717-723):

In general, the only curative option currently available for CPFE is lung transplantation, shown to be feasible and coupled with favourable outcomes in CPFE patients [186]. However, it was reported that compared to the IPF patients, those with CPFE are more likely to develop primary graft dysfunction (grade 3), acute cellular regression (grade ≥2), and chronic allograft dysfunction after lung transplantation. Nevertheless, despite these distinctive outcomes, the study reported comparable survival of IPF and CPFE patients after transplantation [187].

  1. Tathagat, N.; Archer, K.M.; Abuzar, A.A.; Ashley, V.F.; Zhuo, L.; David, B.E.; Francisco, A.; Mathew, T. Outcomes of Lung Transplantation in Patients With Combined Pulmonary Fibrosis and Emphysema: A Single-Center Experience. Transplantation Proceedings 2023, doi:https://doi.org/10.1016/j.transproceed.2023.01.010.
  2. Takahashi, T.; Terada, Y.; Pasque, M.K.; Liu, J.; Byers, D.E.; Witt, C.A.; Nava, R.G.; Puri, V.; Kozower, B.D.; Meyers, B.F.; et al. Clinical Features and Outcomes of Combined Pulmonary Fibrosis and Emphysema After Lung Transplantation. Chest 2021, 160, 1743-1750, doi:10.1016/j.chest.2021.06.036.

Reviewer 2 Report

I read with great interest the paper by Gredic et al (cells-2221555) that summarized current knowledge about clinical attributes and manifestations of CPFE. This study focused on pathophysiological and pathohistological lung phenomena and suspected etiological factors of CPFE. Moreover, since there is a paucity of preclinical research performed for this particular lung pathology, this study reviewed existing animal studies, and provided suggestions for the development of additional in vivo models of CPFE syndrome.

I think it is written very well and I see no particular areas that need correction. 

Author Response

Reviewer 2.

Q1: I read with great interest the paper by Gredic et al (cells-2221555) that summarized current knowledge about clinical attributes and manifestations of CPFE. This study focused on pathophysiological and pathohistological lung phenomena and suspected etiological factors of CPFE. Moreover, since there is a paucity of preclinical research performed for this particular lung pathology, this study reviewed existing animal studies, and provided suggestions for the development of additional in vivo models of CPFE syndrome.

I think it is written very well and I see no particular areas that need correction. 

R1: We are grateful to the reviewer for the nice overview and for the full support of our manuscript.

Reviewer 3 Report

The review entitled "Combined pulmonary fibrosis and emphysema: when Scylla and Charybdis ally" is  of some interest however many critical points are present and the manuscript should be accurately reviewed.

To many abbreviations are present in the text: a list of abbreviation should be reported.

The introduction should be rewritten, some sentences are not clear. 

In the text the references are not correctly reported. e.g.: Cottin in 2005 instread of Cottin et al., in 2005 [5] coined... 
The paragraph ..Pathogenesis should be before the paragraph Pathophysiology, moreover it is too long and ripetitive.

Table 1: Cass et al., Single intratrachel dose of... What.. also in the line of Kulshres et al.. There are many trivial mistakes and the information reported in the table 1 and 2  should be synthesized.

The reported studies should be discussed the conclusion rewritten in accordingly.

The references should be reported in accord to the editorial rules of the journal

Author Response

Reviewer 3.

The review entitled "Combined pulmonary fibrosis and emphysema: when Scylla and Charybdis ally" is  of some interest however many critical points are present and the manuscript should be accurately reviewed.

Q1: To many abbreviations are present in the text: a list of abbreviation should be reported.

R1: We agree with the reviewer that our manuscript contains many abbreviations and we did our best to explain them when they first appear in the text. However, the journal provides the Microsoft Word template that we were expected to follow with regard to structure and formatting of the paper. Thus, we omitted the list of abbreviations in order to comply with the style of the journal.

Q2: The introduction should be rewritten, some sentences are not clear. 

R2: We thank reviewer for this suggestion. We carefully checked the introduction and simplified the sentences where needed.

Q3: In the text the references are not correctly reported. e.g.: Cottin in 2005 instread of Cottin et al., in 2005 [5] coined... 

R3: We thank the reviewer for drawing our attention to this problem. We thoroughly checked the manuscript and corrected the reporting of references whenever needed.

Q4: The paragraph ..Pathogenesis should be before the paragraph Pathophysiology, moreover it is too long and ripetitive.

R4: We agree with the reviewer that in such manuscripts pathogenesis is usually described before pathophysiology. However, in this case the topic is relatively novel disease entity with very complex pathophysiological features, which are not necessarily a common knowledge among all the readers in the field of emphysema or lung fibrosis. Therefore, we decided to describe the pathophysiology before proceeding to possible pathogenic factors. For the same reason - relative novelty of the CPFE and scarce information about the pathogenesis of the disease, we decided to thoroughly cover the pathogenesis and include all the aetiologies we encountered, even if it resulted in a significant volume of text.

Q5: Table 1: Cass et al., Single intratrachel dose of... What.. also in the line of Kulshres et al.. There are many trivial mistakes and the information reported in the table 1 and 2  should be synthesized.

R5: In order to use the limited space available in the Table in the best possible way, the type of treatment is given in the headline (e.g. smoke exposure, bleomycin injury, MHV injection etc.). Therefore, “single intratracheal dose” in the example refers to bleomycin, since it is in the column “bleomycin injury”. We hope that this clarifies the confusion about the information presented in the tables.  In accordance with reviewers’ comment, we merged the tables 1 and 2.

Q6: The reported studies should be discussed the conclusion rewritten in accordingly.

R6: We agree with the reviewer that discussion of the reported studies is desirable in review papers. We indeed attempted to discuss the reported findings, whenever there was sufficient data from the literature to form a larger, general picture. However, it is unusual to have a separate chapter for the discussion in a review paper, and here it would be hardly manageable, as we covered quite a wide range of topics in this review. Therefore, we decided to discuss them in the sub-chapters, as they appear (for instance, we discussed findings from the animal models in the respective chapter and where possible, compared them to the previously described pathophysiological changes in human lungs.) For the same reason, and also the fact that the field is still quite young and insufficiently investigated, we don’t feel that is possible to give stronger and more definite conclusion than we already did.

Q7: The references should be reported in accord to the editorial rules of the journal

R7: For the reporting of references, we used EndNote program and the formatting style downloaded from the MDPI website (https://www.mdpi.com/authors/references). We are aware that the program is not perfect and occasionally introduces some mistakes, however in our experience those are corrected later in the editing process, when all the revisions and changes of text are done.

Reviewer 4 Report

Extensive and informative review. 

I have only one major revision to suggest:

- Although this is a narrative review, a paragraph of methodology need to be added after the introduction. This paragraph must contain the keywords used and how many authors performed the literature screening. Inclusion and exclusion criteria for literature search. The database searched. And eventual strengths and weaknesses of this search. 

Author Response

Reviewer 4.

Q1: Extensive and informative review. 

I have only one major revision to suggest:

- Although this is a narrative review, a paragraph of methodology need to be added after the introduction. This paragraph must contain the keywords used and how many authors performed the literature screening. Inclusion and exclusion criteria for literature search. The database searched. And eventual strengths and weaknesses of this search. 

R1: We thank the reviewer for this important comment. In agreement, we have added a paragraph of methodology in the revised manuscript version (marked in red, page 2, lines 60-78), as follows: “The relevant literature sources analyzed in our manuscript were searched using the PubMed and Google Scholar. The following keywords, alone or in different combinations, were used for the literature screening: pulmonary fibrosis, lung fibrosis, emphysema, chronic obstructive pulmonary disease, combined pulmonary fibrosis and emphysema, usual interstitial pneumonia, smoking-related interstitial lung disease, idiopathic pulmonary fibrosis, interstitial pneumonia with emphysema, emphysema with fibrosis, diffuse interstitial pneumonia with emphysema, pulmonary hypertension, computed tomography, fibrotic interstitial lung disease, fibrotic non-specific interstitial pneumonia, histology, histopathology, fibroblast foci, honeycombing, honeycomb lesions, pulmonary function, connective tissue disease, bleomycin, cigarette smoke exposure, smoke exposure, mice, animal models and transgenic mouse model.  In general, the inclusion criteria were the following: 1) articles written in English language, 2) papers with appropriate study design, and accurate, objective and reliable data presentation, interpretation and conclusions, 3) research and review articles that comprehensively covered the relevant topic and 4) papers that provided the crucial information. The articles written in other languages and those who did not meet the inclusion criteria were excluded from the further analysis. This way of the literature search is characterized by one important weakness, such as neglect and ignorance of the data/findings in the papers that are not published in English language.” Finally, the authors that were involved in the literature search were already identified in the authors´ contribution (page: 19).

Round 2

Reviewer 3 Report

The paper could be accepted after a minor revision.

Specific comments:

1. Acronyms in paragraphs' title should not be reported. Paragraph 3.2.2 CT in extensive.

2. A paragraph with the list of abbreviations should be reported

Author Response

Reviewer 3.

Q1. The paper could be accepted after a minor revision.

Specific comments:

  1. Acronyms in paragraphs' title should not be reported. Paragraph 3.2.2 CT in extensive.
  2. A paragraph with the list of abbreviations should be reported

R1: We thank the reviewer for support of our manuscript. In agreement with the reviewer, we have corrected the issue about the acronyms in paragraphs´ titles (pages: 3-4, marked in red) and we have added a paragraph with the list of abbreviations (pages: 19-20, marked in red).